# The Amount of Light to Vigorous Physical Activity (Met’s-Hours/Day) in Children with and without Down Syndrome Attending Elementary School in Japan

**DOI:** 10.3390/ijerph20021293

**Published:** 2023-01-11

**Authors:** Erika Yamanaka, Takayo Inayama, Kanzo Okazaki, Tsubasa Nakada, Michio Kojima, Ichiro Kita, Kazunori Ohkawara

**Affiliations:** 1Graduate School of Human Health Sciences, Tokyo Metropolitan University, Tokyo 192-0397, Japan; 2Department of Food and Health Sciences, Faculty of Health, and Human Development, The University of Nagano, Nagano 380-8525, Japan; 3Department of Human Science, Faculty of Liberal Arts, Tohoku Gakuin University, Sendai 981-3193, Japan; 4Graduate School of Informatics and Engineering, The University of Electro-Communications, Tokyo 182-8585, Japan; 5Faculty of Human Sciences, University of Tsukuba, Tokyo 112-0012, Japan

**Keywords:** Down syndrome, Japanese elementary school students, physical activity, light physical activity, moderate-to-vigorous physical activity, light-to-vigorous physical activity

## Abstract

Children with Down syndrome (DS) have physical characteristics such as hypotonus of the musculature. Therefore, their attainment rate of physical activity guidelines is low, and guidelines alone may not be sufficient in assessing the amount of physical activity in children with DS. Compared with normal children (NC) of the same grade, light physical activity (LPA) must be considered while assessing physical activity of children with DS, owing to muscle hypotonia. This study included 69 children with DS and 68 NC in grades 4–6 attending elementary school in Japan. The measurements for physical characteristics included age, height, weight, and body mass index. Physical activity was measured using a triaxial accelerometer, which indicated physical activity volume. Children with DS had less moderate-to-vigorous physical activity duration (DS: 53.1 min/day, NC: 65.0 min/day; *p* < 0.001) but significantly longer LPA duration (DS: 376.4 min/day, NC: 287.7 min/day; *p* < 0.001) than NC. Conversely, the amount of light to vigorous physical activity (Met’s-hours/day) was greater in children with DS (DS: 16.0 Met’s-hours/day, NC: 14.4 Met’s-hours/day; *p* = 0.037). In children with DS with muscular hypotonia, vigorous physical activity is challenging, but LPA is feasible. Developing and validating educational programs that promote physical activity with intensity level depending on individual’s physical characteristic are warranted.

## 1. Introduction

Down syndrome (DS) is a congenital disorder caused by trisomy of chromosome 21. It is characterized by short stature, decreased muscle tone, and mild mental retardation [1,2]. Obesity is one of the health challenges for children and adults with DS [3,4,5]. The risk of obesity in adulthood is becoming more serious [3,4] as the life expectancy of people with DS increases [2,6]. Promoting health by preventing and reducing obesity in childhood is imperative.

Physical activity is essential to reduce the risk of obesity [5,7,8]. According to the Physical Activity and Sedentary Behavior Guidelines (2020), the World Health Organization (WHO) recommends that children and adolescents (ages 5–17) with and without disabilities engage in at least 60 min of moderate to vigorous physical activity (MVPA), alongside reducing sedentary behavior (SB) [9].

It is desirable for the children with DS to develop a healthy lifestyle to prevent lifelong obesity. MVPA ≥ 60 min/day or more of physical activity is reported beneficial in children with DS [10,11,12]. However, reports suggest that achieving more than 60 min/day of MVPA is difficult for children with DS [11,13,14,15,16,17]. In contrast, the studies by Matute-Llorente et al. [11,15], which compared the amount of physical activity in adolescents with and without DS, found lesser MVPA duration but more light physical activity (LPA) duration. If moderate physical activity (MPA) and vigorous physical activity (VPA) is low, but LPA duration is high, the total amount of physical activity per day (Met’s-hours) may not be small. However, the findings of Matute-Llorente et al. [11,15], owing to a small sample size and probable selection bias, require further research.

Children with DS generally have difficulty initiating and sustaining VPA. Therefore, attention is required in applying the guidelines [9], which use the intensity of activity for normal children (NC) as an indicator. In our previous study, about 50% of children with DS attending elementary school in Japan did not achieve MVPA of more than 60 min/day [17]. However, we thought that the LPA duration could be high, as seen in the studies of Matute-Llorente et al. [11,15]. If so, it may be desirable to evaluate physical activity in children with DS not only in terms of MVPA times, but also in terms of physical activity, amount including LPA, activity intensity (Met’s), and × duration (hours). We assessed the amount of daily physical activity in children with DS and NC attending elementary schools in Japan using a triaxial accelerometer. The purpose of this study was to compare the physical activity of children with DS having low muscle tone with that of NC of the same grade level and to determine the need to consider LPA when assessing physical activity in children with DS. Furthermore, we aimed to clarify the need to develop educational programs to promote physical activity that considers the physical characteristics of children with DS, which differ from those of NC.

## 2. Materials and Methods

### 2.1. Research Design

This is an analytical observational study with comparisons and controls.

### 2.2. Participants

Our study included 69 children with DS and 68 NC in grades 4–6 attending elementary school in Japan. The method of selecting the subjects was based on the fact that the subjects of the NC data were in grades 4–6, and accordingly, only the same grades were used for the children with DS data. The physical activity data for children with DS were obtained from children belonging to the Japan Down Syndrome Society, collected from 2016 to 2018 [17]. The physical activity data for NC were obtained from children attending “A” Elementary School in Tokyo, Japan, as measured in 2017 in a previous report [18]. The percentages of the number of children with DS and NC in terms of grades and sex is approximately the same. The survey methods used are similar to those of previous reports [17,18].

### 2.3. Measurement Items and Methods

We used similar items and methods to assess attributes, physical characteristics, and physical activity including sex, age, grade, height, and weight, which were obtained from a questionnaire completed by the parents in the case of children with DS and by the individual in the case of NC [17,18]. Body mass index was calculated as weight in kilograms divided by height in meters squared. Overweight/obesity was determined per the Japanese weight cut-offs on the basis of national reference data for Japanese children [19].

Physical activity was calculated using a validated [20,21] triaxial accelerometer (Active style Pro HJA-750C, Omron Healthcare, Inc., Kyoto, Japan) as well as using the child estimation formula [22]. The triaxial accelerometer was worn on a belt with an activity meter cover by the children with DS and on a special clip by the NC, on the waist for one consecutive week, including weekends. Exceptions to this were during activities such as swimming, changing clothes, and bathing, as well as when sleeping.

Data were analyzed only if available for more than three weekdays and more than one weekend day, wherein the device was worn for at least 600 min/day (and removed for less than 120 min). These accelerometry methods including non-wear time and valid wearing minutes and days were employed in accordance with a previous study [23]. The analysis was conducted from 7:00 a.m. to 9:00 p.m. Physical activity was assessed in terms of time (minutes/day) and amount (Met’s-hour/day) by intensity. Physical activity intensity was classified into SB (PA < 1.6 Met’s), LPA (1.6 Met’s ≤ PA < 3.0 Met’s), MPA (3.0 Met’s ≤ PA < 6.0 Met’s), VPA (6.0 Met’s ≤ PA), MVPA (3.0 Met’s ≤ PA), and LMVPA (1.6 Met’s ≤ PA).

### 2.4. Statistical Analysis

Physical characteristics were analyzed by sex, with means (95% confidence intervals) for interval scales, numbers, and percentages (%) for obesity. Physical activity (minutes and Met’s-hour) by activity intensity was tabulated using macros provided free of charge by the Physical Activity Research Platform [24]. Average values were calculated by weighting for 5 weekdays and 2 weekend days (weighted data = ((average for weekdays × 5 days) + (average for weekend days × 2))/7) (95% confidence interval).

Tests for differences between children with DS and NC were based on Student’s *t*-test when the interval scales were both normally and equally distributed, Welch’s *t*-test when not equally distributed, and Mann–Whitney’s test when the interval scale was not normally distributed. Tests for differences in the duration of physical activity (min/day) and the amount of physical activity (Met’s-hour/day) were determined with wearing time and sex as covariates. The amount of physical activity was also checked for normality, but some were non-normally distributed. Because of the difference in wearing time between the two groups, the analysis of covariance method was used, which allows for adjustment. The test of proportions was based on the test of χ-squared independence. Statistical analysis was performed using IBM SPSS Statistics (version 23.0; IBM Co., Tokyo, Japan), with a statistical significance level of 5%.

This study was approved by the Research Safety and Ethics Committee of Tokyo Metropolitan University (H29-72), the Ethics Committee of the University of Nagano (E18-1), and the Ethics Committee of Tohoku Gakuin University (2017R001).

## 3. Results

The physical characteristics of the children with DS and NC are shown in Table 1. The children with DS had a significantly lower height and weight than the NC according to sex and grade category (*p* < 0.05). The percentage of obesity, as determined by the BMI percentile values, did not differ significantly between the children with DS and NC. However, in the determination of obesity by standard weight by sex and height in Japan, the proportion of obesity in children with DS was significantly higher than that in NC in both girls and boys (*p* = 0.001 and *p* = 0.025, respectively).

Table 2 describes activity duration by physical activity intensity for children with DS and NC. The percentage of children performing 60 min/day or more of MVPA was 33.3% for DS and 48.5% for NC, although there was no significant difference. LPA durations were children with DS: 376 min/day and NC: 288 min/day, being significantly more in children with DS than in NC (*p* < 0.001). MPA duration was significantly less for children with DS: 50.3 min/day compared with NC: 56.7 min/day; VPA duration was significantly less for children with DS: 2.8 min/day compared with NC: 8.3 min/day (*p* = 0.011 and *p* < 0.001, respectively).

The amount of physical activity (Met’s-hour/day) by intensity for children with DS and NC is shown in Table 3. The children with DS had lower amounts of MPA, VPA, and MVPA (all *p* < 0.001) but higher amounts of LPA and LMVPA (*p* < 0.001 and *p* = 0.037, respectively) than the NC.

## 4. Discussion

In the present study, the amount of physical activity of children with DS was compared with that of NC, and the MVPA (minutes/day) was shorter, but the LPA (minutes/day) was longer and the amount of LMVPA (Met’s-hours/day) was equal or higher.

Children with DS have a slower growth rate compared to NC [2,25]. Notably, a high percentage of those with children with DS are obese [5,8,26,27]. Similarly, here, children with DS were shorter, underweight, and had a higher percentage of obesity compared to NC of the same age group. Given the slower growth rate, it is challenging to compare children with DS and NC of the same age.

The results of this study, in which the percentage of those children with DS attaining the guideline (MVPA of 60 min/day or more) was approximately 30%, are consistent with the range of 0% to 43% for children with DS, as reported by Fox et al. [12]. The percentage of NC attaining the guideline is 15% for elementary school children in England [28], and the United States NHNES (The United States National Health and Nutrition Examination Survey) guideline attainment for boys aged 6 to 11 years is 46% and 22% for girls of the same age range [29]. The percentage of elementary school students living in Japan achieving the guidelines was 28% in a previous study [30] and approximately 50% in our study. The reason for the wide range of achievement rates among the previous studies is due to the varying activity meters used for measurement and different cut-off points for activity intensity. Therefore, although it is difficult to compare the results of this study with those of previous studies, it may be concluded from the results of this study, which evaluated the same model and criteria, that the percentage of children with DS who achieved the guidelines may be lower than that of NC. Contrarily, a 50% rate of guidelines attainment of the NC means that programs promoting physical activity are cardinal for all children, regardless of disabilities.

The evaluation of intensity revealed that children with DS were less proficient than NC in performing MPA and VPA activities and sustaining those activities but were able to attain LPA durations. Whitt-Glover et al. compared siblings with DS and NC living in the USA in the same family environment [26] and found that children with DS (*n* = 28, mean age 6.6 years) had LPA, MPA, and VPA of 335.2 ± 105.6 min/day, 153.1 ± 56.4 min/day, and 49.5 ± 29.9 min/day, respectively, whereas NC siblings (*n* = 30, mean age 7.1 years) had 303.2 ± 114.3 min/day, 154.6 ± 57.2 min/day, and 68.6 ± 37.0 min/day, respectively, indicating that DS siblings had significantly less VPA times. Whitt-Glover et al. [26] stated that increasing VPA may be beneficial for obesity prevention and health promotion, as children with DS had a higher BMI. Matute-Llorente et al. also examined the relationship between physical activity and bone mineral density [11] and cardiopulmonary function [15] in children and adolescents with DS and NC living in Spain. Additionally, a study [11] of children and adolescents with DS (*n* = 19, mean age 14.7 years) and NC (*n* = 14, mean age 13.2 years) showed that children and adolescents with DS had significantly more LPA duration (DS: 243.5 ± 39.5 min/day vs. NC: 181.1 ± 39.8 min/day, *p* < 0.05) and significantly less VPA time (DS: 7.4 ± 8.2 min/day vs. NC: 15.8 ± 8.4 min/day, *p* < 0.05). Another report [15] suggested that children and adolescents with DS (*n* = 27, mean age 16 years) had significantly more LPA duration and significantly less MPA and VPA duration than NC (*n* = 15, mean age 13.6 years), both having *p* < 0.05. In this report, the authors stated that spending more time doing MPA may promote better cardiorespiratory fitness in patients with DS, since there was a correlation between VO2max, HRmax, and MPA times in children and adolescents with DS. Additionally, the subjects in this study also had fewer VPA sessions and more LPA sessions. The United States, Spain, and Japan have different elementary school curriculums. In Japan, for example, most DS patients attend special schools for children with disabilities only or special classes for children with disabilities (support class) in a local elementary school, providing education based on the respective curriculum [31]. Although it is necessary to consider the differences in the school environment of the children, the existing prevalence of LPA duration, across various studies, can be attributed to children with DS. These requisites the need to consider not only the amount of physical activity (in minutes) in terms of intensity but also the amount of physical activity (in minutes) per day, including LPA.

The higher proportion of LMVPA in children with DS from the study (Met’s-hour/day) indicates a higher impact of LPA on total physical activity. MVPA is proven to not only prevent obesity in children with DS but also reduces functional activity impairment due to decreased physical fitness [10,11,12]. Physical Activity and Sedentary Behavior Guidelines [9] recommend performing MVPA at least 60 min/day and reducing time spent sedentary to promote health in children and adolescents with disabilities. However, for children with physical impairments such as hypotonus in DS, calculating physical activity as total amount per day may be more appropriate, instead of the intensity. Further research is warranted to investigate the benefits of LPA.

This study has several limitations. First, considering the limited number and disparate locale of residency of the participants, generalizing the results of this study is challenging, as the child’s physical activity is also influenced by the environment, including family and community. Second, only the data available from 7:00 to 21:00 were used in the analysis, making it difficult to observe children with alternate lifestyles, such as those that stay up late, which is typical in upper grades of elementary school. Third, there were differences in wearing time. Therefore, we analyzed wearing time as a covariate. Fourth, we excluded data from days consisting of an unattached period of more than two hours, regardless that the unattached time was spent swimming or playing in water. This may underestimate physical activity. Last, this study compared children with DS and NC studying in the same grade, although children with DS grow and develop relatively slowly compared with NC. Whether differences in growth and development affect physical activity may be an issue to be examined in the future.

Despite these limitations, the present study, for the first time, shows that compared with NC, physical activity in children with DS needs to be examined not only in times per intensity level, but also in physical activity per day (minutes/day and Met’s-hour/day), including LPA. On the other hand, even though the total physical activity of children with DS in this study was higher than that of NC, a higher percentage of them were obese. Therefore, the issue that needs to be considered is the extent to which the maintenance and increase in total physical activity due to increased LPA has an impact on the health maintenance of children with DS.

## 5. Conclusions

In children with DS with hypotonus of the musculature, maintaining total daily physical activity may be possible by extending the LPA duration. Consequently, LPA should also be considered in the assessment of physical activity per day for children with DS. Furthermore, it will be necessary to develop and validate educational programs for promoting physical activity that acknowledge intensity of the activities corresponding to physical characteristics.

## Figures and Tables

**Table 1 ijerph-20-01293-t001:** Study participants’ attributes and physical characteristics.

Variables		DS ^1^ *n* = 69 ^2^				NC ^1^ *n* = 68 ^3^				DS vs. NC	
		Boys, *n* = 35		Girls, *n* = 34		Boys, *n* = 31		Girls, *n* = 37		Boys	Girls
		Mean	(95% CI)	Mean	(95% CI)	Mean	(95% CI)	Mean	(95% CI)	*p* ^5^	*p* ^5^
Age	grade	10.6	(10.3–10.9)	10.4	(10.1–10.7)	10.9	(10.6–11.2)	10.9	(10.6–11.3)	0.281	0.020
Height (cm)	4th	123.6	(120.3–127.0)	122.1	(116.7–127.5)	139.7	(136.1–143.2)	138.8	(134.5–143.0)	<0.001	<0.001
	5th	131.8	(129.1–134.5)	128.2	(125.6–130.8)	148.4	(143.1–153.7)	144.8	(140.9–148.6)	<0.001	<0.001
	6th	137.5	(132.7–142.3)	131.8	(126.7–136.9)	151.2	(145.6–156.7)	155.6	(151.8–159.4)	<0.001	<0.001
Weight (kg)	4th	26.9	(22.5–31.4)	25.7	(21.9–29.5)	35.3	(30.7–39.9)	30.2	(28.4–32.1)	0.012	0.030
	5th	31.9	(29.5–34.4)	28.5	(26.2–30.8)	37.6	(33.2–41.9)	35.0	(31.4–38.6)	0.014	0.002
	6th	37.6	(31.6–43.5)	33.8	(28.3–39.3)	40.8	(35.9–45.7)	42.2	(36.6–47.7)	0.014	0.030
BMI (kg/m^2^)	4th	17.5	(15.3–19.7)	17.0	(16.0–18.1)	18.0	(16.2–19.7)	15.7	(15.0–16.4)	0.725	0.021
	5th	18.4	(17.0–19.7)	17.3	(16.3–18.3)	17.0	(15.5–18.6)	16.6	(15.4–17.9)	0.159	0.381
	6th	19.7	(17.2–22.2)	19.4	(16.7–22.2)	17.8	(16.2–19.4)	17.3	(15.4–19.3)	0.159	0.096
Overweight and obesity ^4^	*n* (%)	12	(34.3)	15	(44.1)	2	(6.5)	1	(2.7)	0.006	<0.001

^1^ Children with Down syndrome (DS), normal children (NC). ^2^ The breakdown of children with Down syndrome: 35 boys (9 in 4th grade, 14 in 5th grade, and 12 in 6th grade) and 34 girls (10 in 4th grade, 15 in 5th grade, and 9 in 6th grade). ^3^ The breakdown of normal children was 31 boys (13 in 4th grade, 10 in 5th grade, and 8 in 6th grade) and 37 girls (12 in 4th grade, 12 in 5th grade, and 13 in 6th grade). ^4^ Overweight and obesity was determined per the Japanese weight cut-offs on the basis of national reference data for Japanese children. ^5^ Interval measures were subjected to Student’s *t*-test if the two groups were normally distributed and equally distributed, or Welch’s *t*-test if they were not equally distributed. Nominal scales were subjected to the χ-square independence test.

**Table 2 ijerph-20-01293-t002:** Physical activity time for children with Down syndrome and normal children.

PA		DS ^1^ *n* = 69		NC ^1^ *n* = 68		DS vs. NC
		Mean	(95% CI)	Mean	(95% CI)	*p* ^8^
SB ^2^	min/day	364.6	(351.8–377.5)	412.6	(400.8–424.4)	<0.001
LPA ^3^	min/day	376.4	(365.2–387.7)	287.7	(277.4–298.0)	<0.001
MPA ^4^	min/day	50.3	(46.6–54.1)	56.7	(52.5–60.9)	0.011
VPA ^5^	min/day	2.8	(2.3–3.3)	8.3	(6.7–9.9)	<0.001
MVPA ^6^	min/day	53.1	(49.1–57.2)	65.0	(59.5–70.5)	<0.001
MVPA ≥ 60 min	*n* (%)	23	(33.3)	33	(48.5)	0.070 ^9^
LMVPA ^7^	min/day	429.6	(416.0–443.2)	352.7	(340.7–364.7)	<0.001
Wearing times	min/day	794	(786–802)	765	(758–773)	<0.001

Results are presented as mean (95% CI) for interval scale and as number (%) for MVPA 60 and above. The table shows unadjusted values. ^1^ Children with Down syndrome (DS), normal children (NC). ^2^ SB: Sedentary behavior: PA < 1.6 Met’s. ^3^ LPA: light physical activity: 1.6 Met’s ≤ PA < 3.0 Met’s. ^4^ MPA: moderate physical activity: 3.0 Met’s ≤ PA < 6.0 Met’s. ^5^ VPA: vigorous physical activity: 6.0 Met’s ≤ PA. ^6^ MVPA: moderate-to-vigorous physical activity: 3.0 Met’s ≤ PA. ^7^ LMVPA: light-to-vigorous physical activity: 1.6 Met’s ≤ PA. ^8^ Analysis of covariance with wearing time and sex as covariates. ^9^ Comparison of the percentage of children who wore the MVPA for more than 60 min was by the χ-squared test.

**Table 3 ijerph-20-01293-t003:** The amount of physical activity (Met’s-hour/day) for children with Down syndrome and normal children.

PA		DS ^1^ *n* = 69		NC ^1^ *n* = 68		DS vs. NC
		Mean	(95% CI)	Mean	(95% CI)	*p* ^8^
SB ^2^	Met’s-hour/day	7.8	(7.6–8.1)	8.5	(8.3–8.7)	<0.001
LPA ^3^	Met’s-hour/day	12.6	(12.2–13.0)	9.7	(9.3–10.0)	<0.001
MPA ^4^	Met’s-hour/day	3.1	(2.8–3.3)	3.6	(3.3–3.9)	<0.001
VPA ^5^	Met’s-hour/day	0.3	(0.2–0.5)	1.1	(1.0–1.3)	<0.001
MVPA ^6^	Met’s-hour/day	3.4	(3.1–3.7)	4.7	(4.2–5.2)	<0.001
LMVPA ^7^	Met’s-hour/day	16.0	(15.4–16.6)	14.4	(13.8–15.0)	0.038

Results are presented as mean (95% CI) for interval scale. The table shows unadjusted values. ^1^ Children with Down syndrome (DS), normal children (NC). ^2^ SB: sedentary behavior: PA < 1.6 Met’s. ^3^ LPA: light physical activity: 1.6 Met’s ≤ PA < 3.0 Met’s. ^4^ MPA: moderate physical activity: 3.0 Met’s ≤ PA < 6.0 Met’s. ^5^ VPA: vigorous physical activity: 6.0 Met’s ≤ PA. ^6^ MVPA: moderate-to-vigorous physical activity: 3.0 Met’s ≤ PA. ^7^ LMVPA: light-to-vigorous physical activity: 1.6 Met’s ≤ PA. ^8^ Analysis of covariance with wearing time and sex as covariates.

## Data Availability

The data presented in this study are available on request from the corresponding author.

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
