# Peer review of "The Amount of Light to Vigorous Physical Activity (Met’s-Hours/Day) in Children with and without Down Syndrome Attending Elementary School in Japan"

_ijerph, 2023, doi:10.3390/ijerph20021293_

Round 1

Reviewer 1 Report

The manuscript seems to be valuable as studies of DS by measuring their physical activities. I have some minor comments.

1.         (Abstract, lines 24-25) It might be better to restructure the sentence by removing the first “physical activity”.

2.         (Introduction, lines 69) Could the authors clarify the meaning of “the number of children with LPA might be high”?

3.         (Materials and Methods, lines 83-84) How did the authors recruit the participants? In addition, the title of the reference number 17 indicates children 7-12 years were the target of the previous study, but why this study targeted only 4-6 grades of elementary school children? Could the authors clarify the flow of selecting the study population?

4.         (Discussion, lines 181-182) It is bizarre that the study revealed “LPA should also be considered when assessing physical activity” because it is an opinion of the authors. How about writing the results shown? In addition, please clarify what was newly revealed by this study in this section.

Author Response

Responses to editorial and reviewers’ comments      Many thanks to the reviewers for the helpful comments on our manuscript.  We have revised the manuscript according to your suggestions.  These changes and additions are underlined in the revised document.  In addition, some minor revisions were made without underline to improve grammar and word choice. Given below are our responses to your comments. 

Reviewer #1 (Comments to the Author):

  1. (Abstract, lines 24-25) It might be better to restructure the sentence by removing the first “physical activity”.

<Authors’ response>

Per your suggestion, revised as follows, including reviewer #2's comments.

Before

Children with Down syndrome (DS) have muscle hypotonia, which may cause the physical ac-tivity to be low for moderate-to-vigorous physical activity (MVPA) and high for light physical activity (LPA).

After

Children with Down syndrome (DS) have physical characteristics such as hypotonus of the musculature. Therefore, their attainment rate of physical activity guidelines is low, and guidelines alone may not be sufficient to assess the amount of physical activity in children with DS.

Please see line 24-26

  1. (Introduction, lines 69) Could the authors clarify the meaning of “the number of children with LPA might be high”?

<Authors’ response> The previous sentence has been corrected and the relevant section deleted in response to your comment.

Before

Although our previous study reported that approximately 50% children with DS attend-ing elementary school in Japan achieved at least 60 min/day of MVPA [17], LPA duration may be high, as seen in the studies by Matute-Llorente et al. [11,15]. We considered that the number of children with LPA might be high.

After

In our previous study, about 50% of children with DS attending elementary school in Ja-pan did not achieve MVPA of more than 60 minutes/day [17]. However, we thought that the LPA duration could be high, as seen in the studies of Matute-Llorente et al [11, 15].

Please see line 66-69

  1. (Materials and Methods, lines 83-84) How did the authors recruit the participants? In addition, the title of the reference number 17 indicates children 7-12 years were the target of the previous study, but why this study targeted only 4-6 grades of elementary school children? Could the authors clarify the flow of selecting the study population?

<Authors’ response>

The reason why the age categories of the participants differed from those in the previous paper is because we matched the age categories of the control group (NC group). The text was modified in response to your comment.

Before

Our study included 69 children with DS and 68 NC in grades 4-6 attending elementary school in Japan.

After

Our study included 69 children with DS and 68 NC in grades 4-6 attending elementary school in Japan. The method of selecting the subjects was based on the fact that the subjects of the NC data were in grades 4-6, and accordingly, only the same grades were used for the children with DS data.

Please see line 84-86

  1. (Discussion, lines 181-182) It is bizarre that the study revealed “LPA should also be considered when assessing physical activity” because it is an opinion of the authors. How about writing the results shown? In addition, please clarify what was newly revealed by this study in this section.

<Authors’ response>

The text was modified in response to your comment.

Before

This study revealed that in children with DS with decreased muscle tone, LPA should also be considered when assessing physical activity

After

In the present study, the amount of physical activity of children with DS was compared with that of NC, and the MVPA (minutes/day) was shorter, but the LPA (minutes/day) was longer and the amount of LMVPA (Met's-hours/day) was equal or higher.

Please see line 186-189

Reviewer 2 Report

Overall, the study is sound and informative and the flow of information is exemplary.  However, a few points and comments need to be considered. 

Can the authors explain it a bit further "phys- 24 ical activity to be low for moderate-to-vigorous physical activity (MVPA) and high for light physical 25 activity (LPA)" I find it a little bit confusing. 

line34. In the abstract, the authors mentioned that there are two categories of PA assessed in their study.  LMVPA was not introduced before. Need clarification. 

line 36. Same comment for VPA

line 57-58. "In contrast, a study 57 by Matute-Llorente et al. [11,15]" although you are referring to a single study yet the references cited include two studies. 

The same happened in line 62. 

"Obesity was determined using two methods. The first method used 94 Int. J. Environ. Res. Public Health 2022, 19, x FOR PEER REVIEW 3 of 9 a BMI percentile chart for Japanese children and determined that children were obese if 95 they exceeded the 87th percentile for boys and the 89th percentile for girls" Why the authors did not rely on the categorization of BMI based on WHO. Additionally, the reference cited [19] do not provide accessible information about percentile for boys and girls. 

Why there are two methods for measuring BMI (line 96-97)

"where the device was worn for at least 600 min/day (and removed for less 107 than 120 min)" Could the authors clarify the reason for using these cut off points. 

In the statistical analysis, was the normality of the data tested? 

Looking at table, with most of the PA categories higher in NC, could the difference in height between the two groups play a factor in this variation?

Discussion

"This study revealed that in children with DS with decreased muscle tone" I suggest deleting decreased muscle tone because it was not assessed in the study. 

"Given the slower growth rate, it is challenging to compare children with DS 188 and NC of the same age." May be as a suggestion comparison can be done with the same height. 

line 258 "The first was to examine whether it is possible 258 to increase MVPA for DS" How this study examined this? Alternatively, it can be said that the study shed some light on the abilities of children with DS to do different categories of PA.

Could the authors add a column with the percentage of children achieving different categories of PA in table 2 and table 3

Author Response

Responses to editorial and reviewers’ comments     

Many thanks to the reviewers for the helpful comments on our manuscript.  We have revised the manuscript according to your suggestions.  These changes and additions are underlined in the revised document.  In addition, some minor revisions were made without underline to improve grammar and word choice. Given below are our responses to your comments.

Reviewer #2 (Comments to the Author):

  1. Can the authors explain it a bit further "physical activity to be low for moderate-to-vigorous physical activity (MVPA) and high for light physical activity (LPA)" I find it a little bit confusing.

<Authors’ response>

Per your suggestion, revised as follows, including reviewer #1's comments.

Before

Children with Down syndrome (DS) have muscle hypotonia, which may cause the physical ac-tivity to be low for moderate-to-vigorous physical activity (MVPA) and high for light physical activity (LPA).

After

Children with Down syndrome (DS) have physical characteristics such as hypotonus of the musculature. Therefore, their attainment rate of physical activity guidelines is low, and guidelines alone may not be sufficient to assess the amount of physical activity in children with DS.

Please see line 24-26

  1. In the abstract, the authors mentioned that there are two categories of PA assessed in their study. LMVPA was not introduced before. Need clarification.

<Authors’ response>

The text was modified in response to your comment.

Before

LMVPA

After

light to vigorous physical activity

Please see line 34

  1. line 36. Same comment for VPA

<Authors’ response>

The text was modified in response to your comment.

Before

VPA

After

vigorous physical activity

Please see line 36

  1. line 57-58. "In contrast, a study 57 by Matute-Llorente et al. [11,15]" although you are referring to a single study yet the references cited include two studies.

<Authors’ response>

The text was modified in response to your comment.

Before

a study by Matute-Llorente et al. [11,15]

After

the studies by Matute-Llorente et al. [11,15]

Please see line 57

  1. The same happened in line 62.

 <Authors’ response>

The text was modified in response to your comment.

Before

a study by Matute-Llorente et al. [11,15]

After

the studies by Matute-Llorente et al. [11,15]

Please see line 62

  1. " Obesity was determined using two methods. The first method used a BMI percentile chart for Japanese children and determined that children were obese if they exceeded the 87th percentile for boys and the 89th percentile for girls " Why the authors did not rely on the categorization of BMI based on WHO.

<Authors’ response>

Thank you for pointing this out. We have reviewed the matter again. Since this study is a comparison of Japanese children, we have decided to adopt the Japanese distribution-based body size assessment recommended by the Ministry of Education, Culture, Sports, Science and Technology. In addition, a previous paper (in English) on Japanese children also used this method.

The relative weight was calculated as [body weight (kg) - standard weight for gender, age, and height (kg)] / standard weight (kg)×100 (%)

 standard weight (kg) = a × measured body height (cm) − b

a and b are gender- and age-specific.

The cut-off values for weight categories were defined as follows: overweight/obesity combined ≥ +20%.

Therefore, the text and tables have been changed.

Before

Obesity was determined using two methods. The first method used a BMI percentile chart for Japanese children and determined that children were obese if they exceeded the 87th percentile for boys and the 89th percentile for girls [19]. The second method was to evaluate children with a relative weight of 20 or more as being obese and prone to obesity, using the Japanese standard weight determination method by sex and height [19].

After

Body mass index was calculated as weight in kilograms divided by height in meters squared. Overweight/obesity was determined per the Japanese weight cut-offs based on national reference data for Japanese children [19].

Please see line 97-99

  1. Additionally, the reference cited [19] do not provide accessible information about percentile for boys and girls.

<Authors’ response>

The text was deleted in response to your comment.

Please see line 97-99

  1. Why there are two methods for measuring BMI (line 96-97)

<Authors’ response>

The text was modified in response to your comment.

Please refer to number 6.

  1. "where the device was worn for at least 600 min/day (and removed for less 107 than 120 min)" Could the authors clarify the reason for using these cut off points.

<Authors’ response>

The text was added in response to your comment.

Before

at least 600 min/day (and removed for less than 120 min).

After

at least 600 min/day (and removed for less than 120 min). These accelerometry methods including non-wear time, and valid wearing minutes and days were employed in accordance with a previous study [23].

Please see line 107-109

  1. In the statistical analysis, was the normality of the data tested?

<Authors’ response>

The text was added in response to your comment.

Before

Tests for differences in the duration of physical activity (min/day) and the amount of physical activity (Met’s hour/day) were determined with wearing time and sex as covariates.

After

Tests for differences in the duration of physical activity (min/day) and the amount of physical activity (Met’s-hour/day) were determined with wearing time and sex as covariates. The amount of physical activity was also checked for normality, but some were non-normally distributed. Because of the difference in wearing time between the two groups, the analysis of covariance method was used, which allows for adjustment.

Please see line 127-129

  1. Looking at table, with most of the PA categories higher in NC, could the difference in height between the two groups play a factor in this variation?

<Authors’ response>

As you point out, height may have an impact. However, it is necessary to examine how differences in growth, not just height, affect physical activity, and this is an issue for the future. I have added this matter to the limits of this issue.

The text was modified in response to your comment. The underlined part has been added.

After

Whether differences in growth and development affect physical activity may be an issue to be examined in the future. Whether differences in growth and development affect physical activity may be an issue to be examined in the future.

Please see line 262-263

  1. (Discussion) "This study revealed that in children with DS with decreased muscle tone" I suggest deleting decreased muscle tone because it was not assessed in the study.

<Authors’ response>

The text was modified in response to your comment, including reviewer #1's comments.

Before

This study revealed that in children with DS with decreased muscle tone

After

In the present study, the amount of physical activity of children with DS was com-pared with that of NC, and the MVPA (minutes/day) was shorter, but the LPA (minutes/day) was longer and the amount of LMVPA (Met's-hours/day) was equal or higher.

Please see line 186-189

  1. "Given the slower growth rate, it is challenging to compare children with DS and NC of the same age." May be as a suggestion comparison can be done with the same height.

<Authors’ response>

Please see the answer to comment #11.

Please see line 262-263

  1. line 258 "The first was to examine whether it is possible to increase MVPA for DS" How this study examined this? Alternatively, it can be said that the study shed some light on the abilities of children with DS to do different categories of PA.

<Authors’ response>

The text was deleted and added in response to your comment.

Before1

The first was to examine whether it is possible to increase MVPA for DS. In Japan, a supportive environment for increasing physical activity, such as dance and swimming classes organized by the organizations concerned, is gradually being established. Therefore, there is a need to devise and evaluate educational programs aimed at promoting MVPA for DS.

After1

Deleted.

Please see line 267

Before2

The second issue is of estimating the necessary increase and maintenance of total physical activity due to increased LPA to bring about an impact on the stability of health in patients with DS.

After2

Therefore, the issue that needs to be considered is the extent to which the maintenance and increase in total physical activity due to increased LPA has an impact on the health maintenance of children with DS.

Please see line 269-271

Accordingly, the following sentence from the conclusions has been deleted.

Before3

However, implementation of MVPA duration for DS or whether it can be increased through educational programs is difficult to determine it the scope of this study.

After3

Deleted.

Please see line 278

  1. Could the authors add a column with the percentage of children achieving different categories of PA in table 2 and table 3

<Authors’ response>

The standard times (or amounts) of physical activity by intensity has not been determined except for MVPA. Therefore, it was not possible to add them to the table.